# Passive-Type Radon Monitor Constructed Using a Small Container for Personal Dosimetry

**DOI:** 10.3390/ijerph17165660

**Published:** 2020-08-05

**Authors:** Yuki Tamakuma, Chutima Kranrod, Takahito Suzuki, Yuki Watanabe, Thamaborn Ploykrathok, Ryoju Negami, Eka Djatnika Nugraha, Kazuki Iwaoka, Mirosław Janik, Masahiro Hosoda, Shinji Tokonami

**Affiliations:** 1Institute of Radiation Emergency Medicine, Hirosaki University, 66-1 Honcho, Hirosaki, Aomori 036-8564, Japan; tamakuma@hirosaki-u.ac.jp (Y.T.); kranrodc@hirosaki-u.ac.jp (C.K.); thamaborn.p@outlook.com (T.P.); m_hosoda@hirosaki-u.ac.jp (M.H.); 2Graduate School of Health Sciences, Hirosaki University, 66-1 Honcho, Hirosaki, Aomori 036-8564, Japan; suzuki-takahito@fujielectric.com (T.S.); h20gg204@hirosaki-u.ac.jp (R.N.); h20gg701@hirosaki-u.ac.jp (E.D.N.); 3School of Health Sciences, Hirosaki University, 66-1 Honcho, Hirosaki, Aomori 036-8564, Japan; watanabe.yuki@jaea.go.jp; 4National Institutes for Quantum and Radiological Science and Technology, 4-9-1 Anagawa, Inage, Chiba 263-0024, Japan; iwaoka.kazuki@qst.go.jp (K.I.); janik.miroslaw@qst.go.jp (M.J.)

**Keywords:** passive radon monitor, development, sensitivity, detection limit, air-exchange rate

## Abstract

The International Commission on Radiological Protection (ICRP) recently recommended a new dose conversion factor for radon based on the latest epidemiological studies and dosimetric model. It is important to evaluate an inhalation dose from radon and its progeny. In the present study, a passive radon personal monitor was designed using a small container for storing contact lenses and its performance was evaluated. The conversion factor for radon (^222^Rn), the effect of thoron (^220^Rn) concentration and the air exchange rate were evaluated using the calibration chamber at Hirosaki University. The minimum and maximum detectable radon concentrations were calculated. The conversion factor was evaluated as 2.0 ± 0.3 tracks cm^−2^ per kBq h m^−3^; statistical analyses of results showed no significant effect from thoron concentration. The minimum and maximum detectable radon concentrations were 92 Bq m^−3^ and 231 kBq m^−3^ for a measurement period of three months, respectively. The air exchange rate was estimated to be 0.26 ± 0.16 h^−1^, whose effect on the measured time-integrated radon concentration was small. These results indicate that the monitor could be used as a wearable monitor for radon measurements, especially in places where radon concentrations may be relatively high, such as mines and caves.

## 1. Introduction

Radon is one of the naturally occurring radionuclides, which is well known as the second leading risk factor for lung cancer after tobacco smoking [1]. According to the United Nations Scientific Committee on the Effects of Atomic Radiation (UNSCEAR) 2008 report on the sources and effects of ionizing radiation [2], the world mean value of annual effective dose by natural radiation sources is 2.4 mSv, and half of this dose is attributed to radon (^222^Rn), thoron (^220^Rn), and their progenies. Radon and thoron, which are radioactive noble gases, are normally generated by radioactive decay from radium (^226^Ra and ^224^Ra) in soil, rocks, and building materials. Indoor radon concentrations tend to be higher than outdoor concentrations due to the ventilation rate. Many indoor radon concentration surveys have been carried out in various countries [3,4,5,6,7]. The world mean indoor radon concentration was reported to be 40 Bq m^−3^ [8]. However, radon concentrations in mines and caves are higher than indoor concentrations because radon generated from surrounding rocks and soil can be easily accumulated due to the low ventilation. Recently, it has been reported that mean radon concentrations at several mines and caves were over ~1000 Bq m^−3^ [9,10,11,12].

In 2017, the International Commission on Radiological Protection (ICRP) recommended a new dose conversion factor of 3 mSv per mJ h m^−3^ for radon in buildings and underground mines based on current epidemiological studies and dosimetric model [13,14]. This value corresponds to 17 nSv per Bq h m^−3^, which is almost twice the dose conversion factor of 9 nSv per Bq h m^−3^ given by the UNSCEAR 2006 report on the sources and effects of ionizing radiation [15]. Many countries will approve the new dose conversion factor of ICRP for the occupational environment under the existing situation [16]. Therefore, it is important to evaluate an inhalation dose from radon and its progeny, especially for workers in mines and caves.

Generally, radon concentration is measured using passive-type monitors recording long-term measurements [17]. However, passive-type monitors have an upper limit of detection. The overlaps of tracks made by alpha particles in the measurement at a high radon concentration area lead to the underestimation of radon concentration. Thus, the sensitivity to radon should be lower than that of previous monitors to measure radon concentrations in the prone area. These monitors are large and inconvenient to carry and they are not designed to be carried by workers. Therefore, a passive radon monitor with low sensitivity to radon for measurement in radon-prone areas that is small enough to be carried during the work period is needed for evaluation of inhalation dose. In the present study, such a passive radon personal monitor was designed and its performance was evaluated.

## 2. Materials and Methods

### 2.1. Overview of the Passive Radon Rersonal Monitor

A passive radon personal monitor is shown in Figure 1a. A small container for storing contact lenses was adapted because it is cheap, and easy to obtain and handle. Each container volume was ~3.2 cm^3^. A solid-state track detector (CR-39; BARYOTRAK, Nagase Landauer, Ltd., Tsukuba, Japan), 1.0 × 1.0 cm, was used as the detecting material and one piece of CR-39 was held in place at the top of each container with sticky clay (Figure 1b). Electroconductive materials have been used for the container of previous passive radon monitors to avoid electrostatic phenomena, which may cause a non-uniform deposition of radon progeny [18,19]. Thus, an electroconductive coating material (D-362, Fujikura Kasei Co., Ltd., Tokyo, Japan) was used on the container’s inner wall. Each container was closed by an attached lid; this makes an invisible air gap that prevents thoron from passing into the monitor. The radon concentration ($X_{\mathrm{Rn}}$) was calculated using the following Equation (1):(1)$$X_{\mathrm{Rn}}=\frac{N-N_{0}}{CF_{\mathrm{Rn}}\cdot T}$$
where $X_{\mathrm{Rn}}$ is the mean concentration of radon during the exposure period in kBq m^−3^; N and $N_{0}$ (tracks cm^−2^) are the track density in the measurement and the background, respectively; $CF_{\mathrm{Rn}}$ (tracks cm^−2^ per kBq h m^−3^) is the conversion factor to radon concentration, which is obtained by a calibration experiment; and T is the exposure time in hours.

### 2.2. Conversion Factor to Radon Concentration

To obtain the conversion factor (CF) to radon concentration, this monitor was placed in the radon calibration chamber [20] at the Institute of Radiation Emergency Medicine (IREM) at Hirosaki University. The radon concentrations in the chamber were continuously monitored by a scintillation cell (300A, Pylon Electronics Inc., Ottawa, ON, Canada) with a portable radon monitor (AB-5, Pylon Electronics Inc., Ottawa, ON, Canada). Three different exposure tests were carried out. Five monitors, totaling 10 CR-39 pieces, were exposed for each exposure condition. After exposure tests, CR-39 pieces were taken from the containers and were chemically etched with a 6.0 M NaOH solution at 60 °C for 24 h. The alpha tracks were counted using ImageJ software (National Institutes of Health, Bethesda, MD, USA) with photos of etched CR-39 pieces. Ten reading areas of 0.01 cm^2^, which corresponds to 0.1 cm^2^, were randomly selected to count the number of the tracks. The CFs for each CR-39 were calculated using Equation (1), and then the arithmetic mean of the CF value was evaluated.

### 2.3. Effect of Thoron Interference

To evaluate the effect of thoron concentration, this monitor was placed in the thoron chamber at IREM. Thoron concentrations were continuously monitored with an electrostatic collection radon-thoron monitor (RAD7, Durridge. Co. Inc., Billerica, MA, USA). A grab sampling technique using a scintillation cell with a portable radiation monitor was used for the correction of values measured by the RAD7 [21]. Temperature and relative humidity in the thoron chamber were simultaneously monitored with an environmental monitor (TR-73U, T&D Co., Matsumoto, Japan). After exposure, the alpha tracks were counted in the same way as for radon exposure. The tracks of the exposed group and background were compared using two-sided Student’s *t*-tests. The difference was considered significant for *p* < 0.05. Excel 2016 software (Microsoft, Washington WA, USA) and R version 3.5.2 were used to perform the statistical analyses.

### 2.4. Minimum and Maximum Detectable Radon Concentrations

The minimum detectable radon concentration was evaluated using the method of the International Organization for Standardization (ISO 11929) [22]. The following equation was used for the calculation of the minimum detectable radon concentration:(2)$$y^{*}=\frac{k\sqrt{2n_{0}}}{S\cdot \mathrm{CF}\cdot T}$$
(3)$$y^{\#}=\frac{2y^{*}+\frac{k^{2}}{S\cdot \mathrm{CF}\cdot T}}{1-k^{2}\left\{u_{\mathrm{rel}}^{2}\left(S\right)+u_{\mathrm{rel}}^{2}\left(\mathrm{CF}\right)\right\}}$$
where $y^{*}$ and $y^{\#}$ are the decision threshold and the minimum detectable radon concentration (kBq m^−3^), respectively; *k* is the quantile of the standardized normal distribution for probabilities of 0.95 (i.e., *k* = 1.65); $n_{0}$ is the number of background tracks (tracks); S is the total reading area (cm^2^); CF is the conversion factor for radon concentration (tracks cm^−2^ per kBq h m^−3^); T is the exposure time (h); and $u_{\mathrm{rel}}\left(x\right)$ is the standard relative uncertainty for the parameter x. The standard relative uncertainty for the total reading area $u_{\mathrm{rel}}\left(S\right)$ was assumed to be zero.

In addition, the maximum detectable radon concentration can be calculated using Equation (1) and the maximum track density estimated by an average track diameter measured using an optical microscopy (Model LVS, Kenis Ltd., Japan). The maximum alpha track density *N*_max_ (tracks cm^−2^) can be calculated by:(4)$$N_{\max}=\frac{1}{d_{t}^{2}}$$
where $d_{t}$ is the average alpha track diameter (cm). Then, the maximum detectable concentration $X_{\mathrm{Rn},\;\max}$ in kBq m^−3^ is expressed as:(5)$$X_{\mathrm{Rn},\;\max}=\frac{N_{\max}-N_{0}}{CF_{\mathrm{Rn}}\cdot T}.$$

### 2.5. Air Exchange Rate

The air exchange rate was evaluated using an exposure–degassing–enclosure method reported by Omori et al. [23]. The 20 monitors with CR-39s were exposed in the radon chamber at IREM, where the radon concentration was stabilized. The radon concentrations were monitored every hour by the pulse-type ionizing chamber (AlphaGUARD, PQ2000PRO, Genitron Instruments GmbH, Frankfurt, Germany), which was calibrated by Physilalish Technische Bundesanstalt (Braunschweig, Germany). After the 48 h exposure, the monitors were taken out of the radon chamber and degassed by leaving them in an experimental room where the radon concentration was low. The degassing time was set to be 0, 1, 2, 3, 4, 6, 8, 10, 12 and 16 h, and the radon gas in the monitor rapidly goes out through the invisible air gap to the experimental room in this period. The monitors were enclosed in a radon-proof bag made from polyethylene for 288 h when each degassing time passed. Table 1 shows the average radon concentration, the environmental parameters in the radon chamber, and the experimental room during the experiment. The average radon concentration observed in the environmental room was quite low compared to that measured in the radon chamber, which indicated the existence of radon in the experimental room during the degassing time would be negligible. After the enclosure, the CR-39s were etched in the same way. For the analysis, 40 reading areas in each CR-39 were taken to avoid increasing the deviation of the track density in this study because the expected difference in the track density for each degassing condition was small.

## 3. Results

### 3.1. Conversion Factor to Radon Concentration

Three different exposures were controlled as 132 ± 1, 309 ± 3, and 564 ± 5 kBq h m^−3^. Using these data, Figure 2 shows the relationship between the time-integrated radon concentration and the alpha track density. The CF to radon concentration was found to be 2.0 ± 0.3 tracks cm^−2^ per kBq h m^−3^.

### 3.2. Effect of Thoron Interference

The mean thoron concentration, temperature, and relative humidity were 6785 ± 2122 Bq m^−3^, 25.5 ± 0.2 °C, and 9 ± 4%, respectively. Time-integrated thoron concentration was controlled as 329 ± 6 kBq h m^−3^. As shown in Figure 3, the track densities for background and exposed CR-39 were 43 ± 17 and 47 ± 27 tracks cm^−2^, respectively. As a result of the statistical analysis, no significant difference in the track density was observed between the background and exposed group (*p* = 0.76).

### 3.3. Minimum and Maximum Detectable Radon Concentrations

Parameters used for the calculation of the minimum detectable radon concentration are summarized in Table 2. Notably, the working time of 8 h per day was considered for the calculation; thus, the measurement times were set to be 720 and 1440 h for three and six months, respectively. The minimum detectable concentrations with a measurement interval of three and six months were evaluated as 92 and 46 Bq m^−3^, respectively.

The average alpha track diameter was estimated as 10 μm. Thus, the maximum detectable radon concentrations with three- and six-month measurements were evaluated as 231 and 116 kBq m^−3^, respectively.

### 3.4. Air Exchange Rate

Figure 4 illustrates the alpha track density with a standard deviation against the degassing time. The track density slowly decreased with a degassing time of 0–8 h. Theoretically, the decreasing trend can be fitted by a single exponential function. As a result, the air exchange rate was estimated to be 0.26 ± 0.16 h^−1^. The result indicates that the air in the monitor is almost exchanged after 8 h from the end of exposure.

## 4. Discussion

### 4.1. Conversion Factor to Radon Concentration

The CFs of various passive integrated radon monitors are summarized in Table 3. Many kinds of passive integrated radon monitors have been developed by different groups, and the CFs are different for each one [18,24,25,26,27]. The CF for the present monitor was almost same or lower compared with the factors for other monitors. This might be due to the difference in the type of solid-state nuclear track detector (SSNTD) used in the monitors. The CFs shown in previous papers might have been evaluated using a different type of SSNTD that has a lower sensitivity than the CR-39 used in the present study. Thus, the CF for RADUET, widely used for a large-scale radon survey, was also evaluated by the same method using two types of CR-39s at the same time to check the difference in sensitivity attributed to the SSNTD. One was provided by Nagase Landauer Ltd., which was used in the present study, and the other was provided by RadoSys Ltd., which was used by Tokonami et al. [18]. The CFs for RADUET evaluated in the present study and the previous paper are summarized in Table 4. The CFs of the CR-39 by Nagase Landauer Ltd. and RadoSys Ltd. were estimated to be 4.4 ± 0.1 and 2.8 ± 0.1 tracks cm^−2^ per kBq h m^−3^, respectively. This result suggested that the sensitivity of SSNTD significantly affects the CF of a passive radon monitor. Notably, the image analysis system and the chemical etching condition might also be significant factors affecting the CF.

McLaughlin and Fitzgerald developed a model to calculate a CF for a cylindrically-shaped passive radon monitor [28]. According to the model, the CFs of the monitors, which have radii of 1.0 and 2.0 cm, corresponding to the present monitor and RADUET, respectively, were estimated to be 1.8 and 3.7 tracks cm^−2^ per kBq h m^−3^, respectively. Compared with the empirical values, the differences in the present monitor and RADUET were 12% and 16%, respectively. Notably, these values calculated by the model include an uncertainty because the model made assumptions that do not reflect the exact geometry and shape of a detector for simplification. The ratio of the two values by the model was calculated as 2.1, whereas that of experimentally evaluated in this study was 2.2. We found a reasonably good agreement between the ratios regardless of the simplified model, indicating that the sensitivity was lower compared with the previous monitor, and the difference in the CFs between the present monitor and RADUET was attributed to the shape of the monitor.

We found that the air gap between the lid and the container prevented thoron from entering the monitor. This result suggested that the effect of thoron concentration is small enough for radon measurements that it could be ignored.

### 4.2. Minimum and Maximum Detectable Radon Concentrations

The results suggested that the present monitor would be useful for radon measurements in places where radon concentrations might be relatively high, such as mines and caves. According to a previous report, radon concentrations in mines and tourist caves have daily and seasonal variations, and its mean annual concentration was around 3500 Bq m^−3^ with a standard deviation of 1833 Bq m^−3^ [12]. There are very few underground mines in Japan so far, but there are some tourist caves; a wearable monitor may be useful as a simple radiation protection instrument for workers.

The long-term indoor radon concentrations are measured using a passive radon monitor, which is put on a shelf or hung on a wall and roof for three or six months. In this case, the minimum detectable radon concentrations are evaluated to be 31 and 15 Bq m^−3^ for three- and six-month measurements, respectively. It has been reported in several foreign countries that their mean indoor radon concentrations are relatively higher than these values [29,30,31,32,33,34]. Therefore, the passive radon monitor developed in this study will be useful not only as a wearable monitor for personal dosimetry due to radon inhalation but also for indoor radon measurements in many countries.

### 4.3. Air Exchange Rate

The trend in the alpha track density against the degassing time in Figure 4 was found to be similar to that reported by Omori et al. [23]. Omori et al. [23] reported that the experimental value was in good agreement with the theoretically calculated value. The air exchange rate of RADUET was previously reported as 0.71 h^−1^, which is higher than that of the present monitor of 0.26 h^−1^ [23]. This indicated that the present monitor has a higher diffusion barrier than RADUET. The monitor is closed by the container’s lid with a screw, which could make the high diffusion barrier, whereas RADUET was designed to be easy to dismantle. As shown in Figure 4, it takes approximately 8 h to completely exchange the air in the monitor, which is why the effect of thoron could be neglected.

To evaluate the effect of air exchange rate on the measurement result when being used as a wearable monitor, the theoretical time-integrated radon concentration in the monitor was calculated considering the working period. It was assumed that workers wear the monitor for a working period of eight hours, and do not wear it at other times. If the radon concentration is 1000 Bq m^−3^, the theoretical time-integrated radon concentration in the monitor is estimated to be 7.6 kBq h m^−3^, whereas the actual value is 8.0 kBq h m^−3^. We found that the difference was estimated to be 5%, which is low compared with other uncertainties such as a counting error. Therefore, the present monitor can be useful as a wearable monitor to evaluate occupational exposure to radon.

## 5. Conclusions

A passive radon personal monitor was designed using a small container for storing contact lenses and its performance was tested. The conversion factor to radon concentration and the effect of thoron concentration on the measurement result were evaluated using the calibration chamber at Hirosaki University. The conversion factor for radon was evaluated as 2.0 ± 0.3 tracks cm^−2^ per kBq h m^−3^ and could be smaller than those with the previous monitors. No significant effect of thoron was observed in the two-sided Student’s *t*-test. The minimum and maximum detectable radon concentrations were estimated as 92 Bq m^−3^ and 231 kBq m^−3^ for a measurement period of three months, respectively. The air exchange rate was evaluated to be 0.26 h^−1^, and the effect could be ignored even if it is used only during the working period. These results indicated that this monitor would be useful as a wearable monitor for radon measurements, especially in caves and mines, and for indoor radon measurements where relatively high radon concentrations are present.

## Figures and Tables

**Figure 1 ijerph-17-05660-f001:**
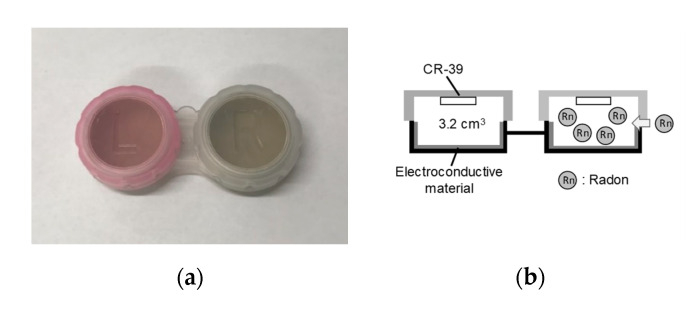
External view of the passive radon personal monitor constructed using a small container for storing contact lenses (**a**) and a schematic representation of the monitor (**b**).

**Figure 2 ijerph-17-05660-f002:**
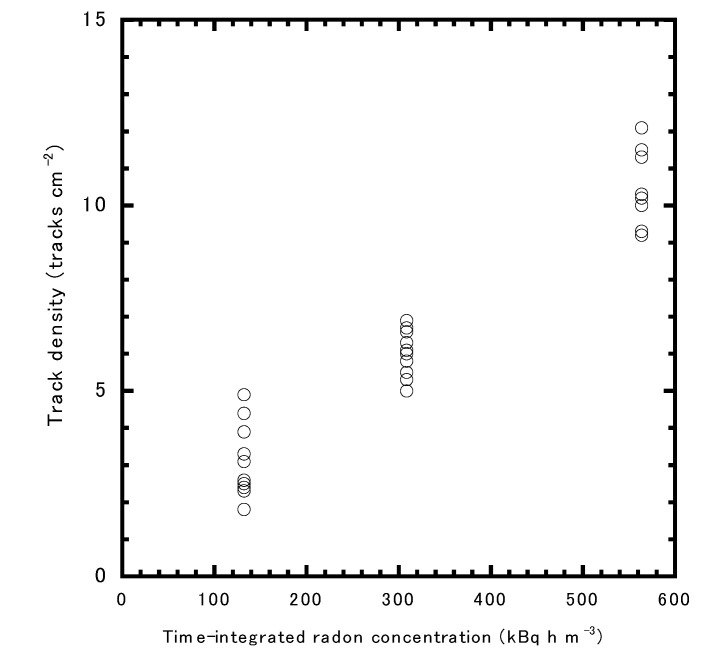
The relationship between the time-integrated radon concentration and alpha track density. Ten CR-39 pieces were exposed for each condition.

**Figure 3 ijerph-17-05660-f003:**
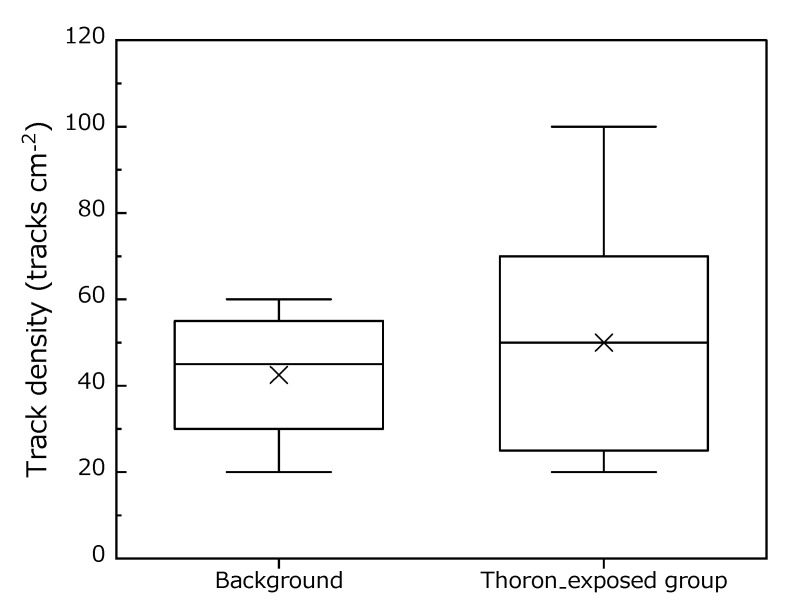
Track densities for the background and thoron-exposed group in unit of tracks cm^−2^. Bottom and top whiskers are the 25th and 75th percentiles, respectively. The lines across the boxes and the cross marks represent the median and the arithmetic mean value, respectively.

**Figure 4 ijerph-17-05660-f004:**
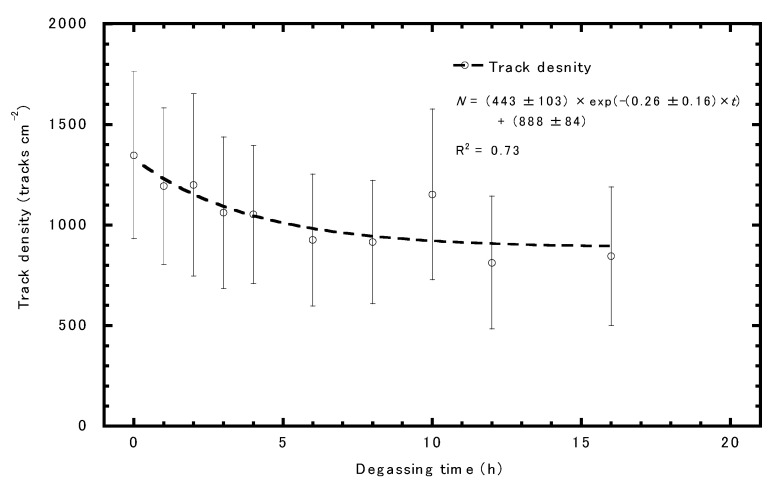
The alpha track density against the degassing time.

**Table 1 ijerph-17-05660-t001:** Radon concentration and environmental parameters in the radon chamber and an experimental room.

Parameter	Radon Chamber		Experimental Room	
	Average	Standard Deviation	Average	Standard Deviation
Radon concentration (Bq m^−3^)	13,300	1603	7.7	3.0
Temperature (°C)	27.7	0.4	26.3	1.5
Relative humidity (%)	50.1	2.5	34.7	6.1
Air pressure (hPa)	1002	2	1003	7

**Table 2 ijerph-17-05660-t002:** Parameters used for the calculation of the minimum detectable radon concentration.

Item	Symbol	Value	Remarks
Quantiles of the standardized normal distribution	*k*	1.65	For probabilities of 0.95
Number of background track	*n_0_*	4.3 tracks	Derived from the background track density of 43 tracks cm^−2^
Total reading area	*S*	0.1 cm^2^	Ten reading areas of 0.01 cm^2^
Conversion factor for radon concentration	*CF*	2.0 tracks cm^−2^ per kBq h m^−3^	
Measurement time	*T*	720 h	3 months
		1440 h	6 months
Relative standard uncertainty for the total reading area	$u_{\mathrm{rel}}\left(S\right)$	0	
Relative standard uncertainty For the conversion factor	$u_{\mathrm{rel}}\left(CF\right)$	0.15	

**Table 3 ijerph-17-05660-t003:** Conversion factors of various passive radon monitors.

Measuring Device	Conversion Factor (Tracks cm^−2^ per kBq h m^−3^)	Reference
Present radon monitor	2.0	
RADUET ^1^	2.3	[18]
KfK monitor ^2^	0.9	[24]
Radtrak	2.8	[25]
NRPB/SSI ^3^	2.2	[26]
Radon-thoron discriminative dosimeter ^1^	1.2	[27]

^1^ The CFs of these monitors were evaluated for those with a low air ventilation rate.^2^ KfK: Kernforschungszentrum Karlsruhe ^3^ NRPB/SSI: National Radiological Protection Board/Statens strålskyddsinstitut.

**Table 4 ijerph-17-05660-t004:** The conversion factors of RADUET evaluated by two types of CR-39s in the present study and in a previous study. RADUETs, which contain both CR-39s, were simultaneously exposed to radon atmosphere for evaluation of the difference in CFs.

Item	Conversion Factor (Tracks cm^−2^ per kBq h m^−3^)	
	Nagase Landauer	RadoSys ^1^
Present study	4.4	2.8
Previous report ^2^	-	2.3

^1^ CR-39s were chemically etched using a 6.25M NaOH solution at 90 °C for 6 h. ^2^. The data were cited from Tokonami et al. [18].

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
