# Peer review of "Passive-Type Radon Monitor Constructed Using a Small Container for Personal Dosimetry"

_ijerph, 2020, doi:10.3390/ijerph17165660_

Round 1

Reviewer 1 Report

The present study addressed a possibility to use of a contact-lenses storage container as radon monitoring. The paper is well organized and easy to be read, so that it would be expected to be accepted after revision based on some minor comments below.

#1: Line 34 (whose effect could be ignored). In this context, the reviewer was not able to understand the meaning of “ignored”. More concrete expression would be expected: Was the air exchange rate negligibly small?; What kind of effect does the air exchange of 0.26+/-0.16 [1/h] have on this monitoring device?

#2: Line 147 (evaluated to be 2.0+/-0.3). Please explain how the authors deduced this value. The reviewer thought the data-fitting approach was taken in Fig 2; If so, a function type used here should also be shown.

#3: Line 149 (Caption of Fig 2). Please mention what was the number of data for each time-integrated radon concentration. Given the description of “39 pieces” at Line 96, n=13 for each can be supposed.

#4: Line 157 (Caption of Fig 3). It must be clearly mentioned that “Exposed group” means “Thoron-exposed group”. The present figure caption has no description of “Thoron”.

#5: Line 125 (2.5. Air exchange rate) and Line 176 (Fig 4). Although the term “degas” is often used, it was not easy to see what the authors actually performed for degassing in this test. The reviewer thought that for this degassing process, the monitors were simply put in the experimental room, keeping their caps closed. If so, how can we interpret the trend of Fig 4 that the track density decreased with time? One may expect that the track density increased with time, since elevated radon concentrations owing to the exposure in the calibration chamber were still kept in the monitors for a while even after putting them in the experimental room.

#6: Line 216 (4.2 Minimum and maximum radon concentration). The reviewer did not think that this title represents the contents appropriately. Please check it accordingly and consider putting a more reasonable title.

Reviewer 2 Report

I have review the paper: “Passive Radon Personal Monitor Made Use of a Small 2 Container for Contact Lenses Storage” by Yuki Tamakuma et al.

The manuscript deals about a passive radon personal monitor designed using a small container for storing contact lenses.

There are some comments highlighting areas of the strength and weakness of this paper.

Strengths:

Introduction: the problem of radon and current regulation is well described

Materials and methods: All the factors that have been taken into account in the development of the passive radon personal monitor are well described: conversion factor, effect of thoron interference, air exchange rate.

Results: Detailed analysis of the results.

Conclusions: The conclusions are supported by the results obtained.

Weaknesses:

I do not understand why it is interesting to use those containers instead of another container with similar characteristics. A more detailed explanation is necessary to justify the usefulness of these containers, material of which it is made, diffusion of radon inside. Is the container really so important that it even appears in the title of the paper? It must be justified, otherwise it must be removed from the title.

In general, the manuscript is interesting. The results are sound and the conclusions are supported by the results. Once my comments are taken into account, I consider it appropriate to publish it in the International Journal of Environmental Research and Public Health.

Reviewer 3 Report

This work developed a passive personal monitor and concluded that it can be used as a wearable monitor for radon measurements, especially in caves and mines and for indoor radon measurements where has relatively high radon concentrations.

The minimum detectable radon concentrations were evaluated as 31 Bq m-3 and 15 Bq m-3 for a measurement period of 2160 hours and 4320 hours. Since the average work time in the caves and mines is 8 hours per day, the time period of 2160h is 9 months but not 3 months and 4320h is 18 months but not 6 months. For indoor monitor, the actually stay time should also be considered. Thus, the sensitivity of the device may be too low for indoor monitoring (average radon concentration is 40 Bq m-3) and too low for some caves and mines with relatively low radon concentrations.

In the discussion part (Line236), “the working period is 8 h for which workers wear the monitor, and they wear it off at other times”, however, the actually situation is not continually 8h, maybe wear on 4 hours, wear off 2 hours and then wear on 4 hours again. So the influence of air exchange rate is more critical for the wearable device. This should be discussed.  

Round 2

Reviewer 3 Report

The manuscript has been improved, I am willing to support the publication.